# Limitations in Validating Derived Soil Water Content from Thermal/Optical Measurements Using the Simplified Triangle Method

**Abba Aliyu Kasim** [1,*] **, Toby Nahum Carlson** [2] **and Haruna Shehu Usman** [3]

[1] Department of Geography and Regional Planning, Federal University Dutsin-Ma, Dutsin-Ma PMB 5001, Katsina State, Nigeria

[2] Department of Meteorology, Penn State University, State College, PA 16801, USA; tnc@psu.edu

[3] School of Continuing Education, Bayero University, Kano PMB 3011, Kano State, Nigeria; hsusman.sce@buk.edu.ng

[*] Correspondence: aakasim@fudutsinma.edu.ng; Tel.: +234-703-611-1018

**Abstract:** We assess the validity of the surface moisture availability parameter ($M_o$) derived from satellite-based optical/thermal measurements using the simplified triangle method. First, we show that $M_o$ values obtained from the simplified triangle method agree closely with those generated from a soil/vegetation/atmosphere/transfer (SVAT) model for scenes over a field site at the Allahabad district, India. Next, we compared $M_o$ values from the simplified triangle method for these same overpass scenes with surface soil water content measured at depths of 5 and 15 cm at this field site. Although a very weak correlation exists between remotely sensed values of $M_o$ for the full scenes and measured soil water content measured at both depths, correlations increasingly improve for the 5 cm samples (but not for the 15 cm samples) as pixels were limited to increasingly smaller vegetation fractions. We conclude that the simplified triangle method would yield reasonable values of $M_o$ and demonstrate good agreement with ground measurements, provided that validation is limited to pixels with little or no vegetation and to soil depths of 5 cm or less.

**Keywords:** simplified triangle method; soil water content; satellite-based optical/thermal measurement; limitation; validating

## 1. Introduction

Surface soil moisture is a regulator that controls the share of rainfall that percolates, runs off, or evaporates from the land surface. Surface soil moisture plays a vital role in apportioning the incoming solar radiation into latent and sensible heat fluxes [1]. Soil moisture is indispensable for the soil–plant growth relationship [2]. In this paper, the remotely sensed surface soil moisture availability ($M_o$) is equated to the fraction of "extractable" soil water content, essentially the ratio of soil water content (SWC) to that of field capacity. Soil moisture varies greatly in time, depth and space. The key soil properties that influence the amount of moisture present in the soil include: soil texture, soil organic matter and soil structure [3,4]. Moreover, the amount of precipitation and the rate of evapotranspiration (ET) play a significant role in drying out of soil, hence soil moisture can be deduced from ET [5].

Initially, geologists applied the technique using remote sensing with optical/thermal measurements to locate mineral deposits while the meteorologists applied the technique so as to estimate surface turbulent energy fluxes and soil moisture availability and the evapotranspiration fraction (EF, defined as the ratio of ET to Rn, the net radiation) [6]. The essential idea behind the use of optical and thermal infrared measurements of surface radiant temperature (Tir) in estimating surface soil moisture is that surface radiant temperature is very sensitive to surface soil moisture, but also to fractional vegetation

cover (Fr) [6]. Similarly, retrieval of soil moisture and vegetation parameters from remotely sensed data has been very promising using microwave remote sensing techniques [7]. Other techniques for inferring soil moisture include: gamma radiation, hyperspectral, thermal and reflected solar radiation, proximal remote techniques such as cosmic-ray neutron and proximal gamma ray techniques [8–11].

Adequate, continuous and reliable information about soil moisture over large areas is limited in many parts of the world despite its multifaceted importance. This is because the conventional point measurements are complex and expensive [12]. Scott et al. [13] opined that there is a need for rapid, less expensive and reliable methods for soil moisture determination since the field-based techniques are cumbersome and expensive and have only limited spatial and temporal coverage. Moreover, many regions lack even the basic meteorological and surface information required for execution of land surface moisture retrieval models based on remote measurements.

Recently, more simplified models have been developed that employ a triangular or trapezoidal geometry of the pixels in Tir/Fr space, where Tir is the radiant surface temperature and Fr is the fractional vegetation cover. These types of models constrain the solution for $M_o$ and EF, having the advantage of requiring no ancillary surface or atmospheric information other than the pixel measurements, while also involving only a simple algebraic framework [14–16].

A question remains as to the accuracy of these simpler models, the simplified triangle method having yet to be fully validated. In order to establish their utility, it is essential that these simpler models be tested against real measurements and against those obtained from more complex models. An object of this paper therefore is to first show that the derived soil surface moisture availability ($M_o$) obtained from the simplified triangle method agrees closely with that same parameter generated from a soil/vegetation/atmosphere/transfer (SVAT) model described by [6,17]. Next, we compare $M_o$ from the simplified triangle method with direct ground measurements of soil water content (SWC) made over the field sites at the Allahabad district, India.

## 2. Materials and Methods

### 2.1. Geographical Description of the Study Site

Allahabad is one of the districts of the Uttar Pradesh state of India. The district is at an elevation of 98 m. It stands at the confluence of the sacred Ganga, Yamuna and the Invisible Saraswati rivers. Latitudinal and longitudinal positions of Allahabad are 25.45°N and 81.84°E respectively [18].

Allahabad is characterized with an annual rainfall of about 1027mm. In the summer, surface temperatures are in the range 40–45 °C while in the winter they range from 2 to 24 °C. This is a typical characteristic of humid subtropical climate according to Köppen [18].

Alluvial soils predominate most of the Allahabad as a result of fluvial actions of river networks in the Ganges system [19]. Moderately shallow and eroded loamy soils are found in the Chaka block (sub-study area) [18,20,21]. Scrub (vegetation dominated by shrubs) is commonly found in the area. Common trees found in the area include: Dhak (*Butea monosprma*), Kakor, (*Ziziphus globerrima*), Aonla (*Emblica ofbicinalis*), Bahera (*Terminalia belerica*), Babul etc. Urban landscape is beautified by artificial vegetation.

The Tehsils (administrative areas) under study are depicted in Figure 1. In situ soil moisture measurements were done in the Chaka block (sub-study area). Some of the plots where sampling took place were under irrigation.

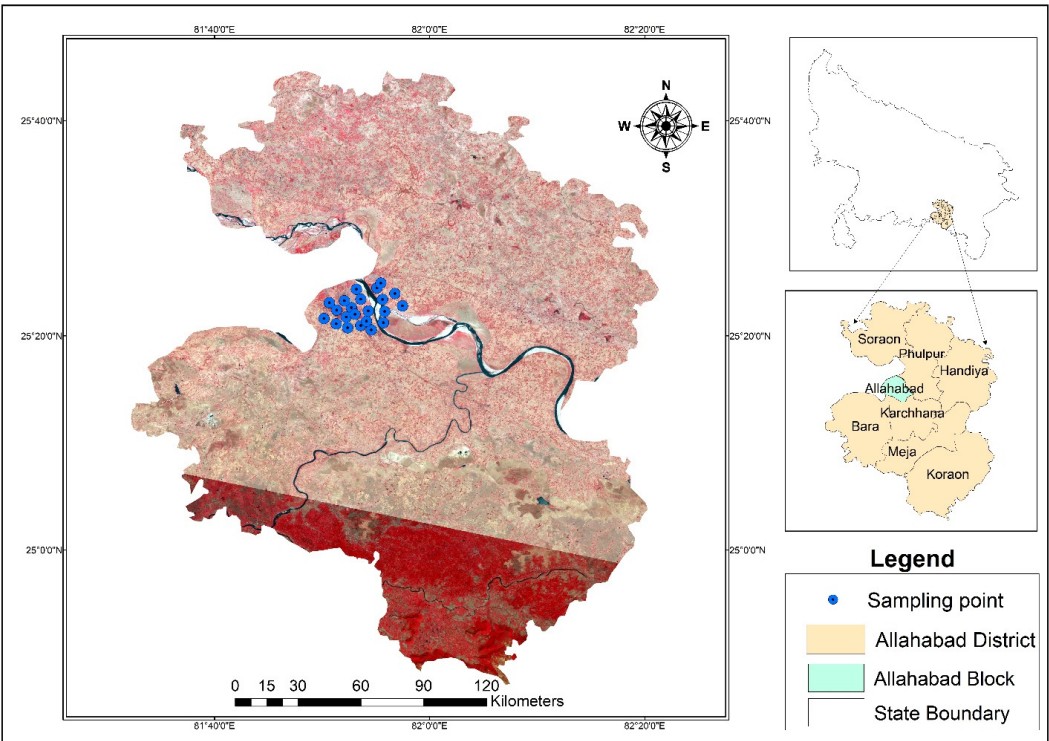

**Figure 1.** Study site and in situ measurement points (main frame: satellite image with sampling points, upper inset: Uttar Pradesh State showing Allahabad and lower inset shows Allahabad district with its blocks).

## 2.2. Satellite Image Processing Operations

The pre-processing and processing operations were carried out within the ERDAS IMAGINE 9.2 environment. Radiometric calibration coefficients of Landsat 7 ETM Plus were used for converting raw digital numbers to radiance according to the Landsat 7 Science Data User's Handbook [22,23] procedures:

$$L_\lambda = \left( \frac{LMAX_\lambda - LMIN_\lambda}{Q_{Calmax} - Q_{calmin}} \right)(Q_{cal} - Q_{calmin}) + LMIN_\lambda \tag{1}$$

where:

$L_\lambda$ = spectral radiance [W/ (m$^2$ sr μm)]; $Q_{cal}$ = quantized calibrated pixel value in DN; $Q_{Calmax}$ = maximum quantized calibrated pixel value (corresponding to $LMAX_\lambda$) in digital numbers (DN); $Q_{calmin}$= minimum quantized calibrated pixel value (corresponding to $LMIN_\lambda$) in DN; $LMAX_\lambda$= spectral at-sensor radiance that is scaled to $Q_{Calmax}$ [W/(m$^2$ sr μm)] in DN; $LMIN_\lambda$= spectral at-sensor radiance that is scaled to $Q_{calmin}$ [W/(m$^2$ sr μm)].

Spectral radiance at the sensor's aperture was converted to planetary top of atmosphere (TOA) reflectance using Equation (2) [22,23]:

$$\rho_\lambda = \frac{\pi.\, L_\lambda.d^2}{ESUN_\lambda.\cos\theta s}, \tag{2}$$

where:

$\rho_\lambda$= unitless planetary reflectance,

$\pi$ = pi [~3.14159],

$L_\lambda$ = spectral radiance at sensor's aperture [W/(m$^2$ sr μm)],

$d^2$ = earth–sun distance in astronomical units [AU],

$ESUN_\lambda$ = mean exo-atmospheric solar irradiance [W/(m$^2$μm)], and

θs = solar zenith angle in degrees.

### 2.3. NDVI and Fr Derivation

NDVI, the normalized difference vegetation index, was calculated using red and near infrared (NIR) spectral bands. The formula (Equation (3)) is as follows:

$$NDVI = \frac{\rho_{nir} - \rho_{red}}{\rho_{nir} + \rho_{red}}, \tag{3}$$

where: $\rho_{nir}$ is near-infrared band reflectance; $\rho_{red}$ is red band reflectance.

NDVI was scaled for the computation of fractional vegetation cover using Equation (4):

$$N^* = \frac{NDVI - NDVI_0}{NDVI_s - NDVI_0} \tag{4}$$

and Fr was derived from Equation (5).

$N^*$ is scaled NDVI (ranges from 0 to 1); $NDVI_0$ and NDVIs are respectively, the minimum and maximum values found in the image once cloud and standing water are removed. The former represents bare soil and the latter represents full vegetation [6,24]. These essential variables can be obtained automatically by computing NDVI layer statistics in the ERDAS IMAGINE 9.2 platform or by simply moving the cursor around the image. In so doing, standing water and cloud pixels are removed. Fractional vegetation cover is then computed from N*:

$$Fr = N^{*2}, \tag{5}$$

where Fr is fractional vegetation cover which ranges from 0 to 1. Fr = 1 means a pixel is filled fully with vegetation while Fr = 0 means a pixel is devoid of vegetation, i.e., it contains just bare soil.

### 2.4. Surface Radiant Temperature Derivation

Surface radiant temperature was derived from the thermal band of the enhanced thematic mapper plus sensor onboard the Landsat 7 platform. The spectral radiance was converted to top of the atmosphere brightness temperature assuming spectral emissivity equal to 1.

$$T = \frac{K2}{ln\left(\frac{K1}{L_\lambda} + 1\right)} \tag{6}$$

where: T = top of the atmosphere brightness temperature in kelvin; K1 = first thermal conversion constant (666.09W/(m$^2$ sr μm)), K2 = second thermal conversion constant (1282.7K); $L_\lambda$ is spectral radiance [W/ (m$^2$ sr μm)] and *ln* is natural logarithm [22,23,25].

### 2.5. Scaled Surface Radiant Temperature Derivation

For the transformation of pixel measurements to surface soil moisture, the surface radiant temperature was used to calculate a scaled temperature using Equation (7) [6].

$$T^* = \left\{\frac{(T_{ir} - T_{min})}{(T_{max} - T_{min})}\right\} \tag{7}$$

where T* = scaled surface radiant temperature; $T_{ir}$, $T_{min}$ and $T_{max}$ are surface radiant temperature, minimum and maximum temperature in the image respectively. $T_{min}$ represents a pixel with lowest temperature value and maximum NDVI value (full vegetation, Fr=1) while $T_{max}$ represents a pixel with highest temperature value and minimum NDVI value (dry bare soil) in the image. These variables are obtained by judicious hand and eye examination of the images, whereby one locates areas of dense vegetation and dry bare surfaces, such as paving. In so doing, T* ranges from 0 to 1, where T*

= 0 corresponds to the temperature of a wet and fully vegetated pixel and T*=1 corresponds to the temperature of a dry pixel devoid of vegetation.

## 2.6. T*- Fr Triangular Space

T* and Fr values were plotted on the X, Y plane, Fr on the y-axis and T* on the x-axis. This arrangement produced triangular shapes on each day under study, as can be seen in Figure 2. These figures clearly show the triangular shape of the pixel envelope in Fr/T* space. Initially, cloud and water pixels distort the shape of T*/Fr triangular space but the bulges were systematically removed. The sharp, slanting right-hand edge of the envelope, referred to as the warm edge, represents the limit of soil dryness for a given value of Fr. The left-hand edge, the cold edge, is typically a vertical line drawn at a constant T* = 0 from Fr = 0 to Fr = 1. After visual inspection of the warm edge, an algorithm (Equation (8)) was also used to fit the warm edge following a procedure described by [12]:

$$T*\text{warm edge} = a + bFr_i,\qquad(8)$$

where *a* and *b* are the intercept and slope of the linear warm edge respectively. In practice, however, fitting a line to the warm edge requires a combination of analytical and trial and error approaches (using one's eye or intelligent manual fit). It should be noted, however, that for a right triangle, such as shown in Figure 2, a warm edge that extends from the vertices T* = 1.0, Fr = 0 to T* = 0, Fr = 1 will have the form T* = 1 − Fr along the warm edge.

## 2.7. Soil Surface Moisture Availability Estimation (M_o)

T*, Fr and coefficients generated from a SVAT model fit to a third-order polynomial equation were used as inputs of Equation (9) for transformation of pixel values to surface moisture availability for a generic case [6]:

$$(M_o) = \sum_{i=0}^{3}\sum_{j=0}^{3} a_{ij}T^{*i}Fr^{j}\qquad(9)$$

where $M_o$ is the moisture availability and $a_{ij}$ are coefficients of the polynomial equation from the SVAT model. The coefficients used to transform the pixel measurements to moisture availability were also adopted from [6]. This equation fits the output of a full SVAT model ($M_o$, EF) for a generic case to a range of input values of Fr and T* [6].

The simplified geometric algorithm based on the triangle shape, developed by [16,26], allows one to calculate $M_o$ from Equation (10). Thus,

$$M_o = 1 - T^*(pixel)/T^*(warm\ edge),\qquad(10)$$

where, as stated above, $T^*_{(warm\ edge)} = (1 - Fr)$; $M_o \leq 1.0$; $\geq 0$.

## 2.8. Satellite Images

Landsat 7 ETM Plus data were utilized. Five cloud free scenes (Path 142; Row 42) were downloaded from the USGS Landsat data repository. Image acquisition dates were 30 January2015; 15 February 2015; 19 March 2015; 20 April 2015 and 06 May 2015.

## 2.9. Ground Reference Measurements

In situ measurements of soil moisture were made in the Chaka block. The gravimetric method was used to determine the soil moisture. A total of 22 ground sampling points were used. Soil moisture measurements were collected at 0–5 cm and 0–15 cm depths during the Landsat 7 ETM plus overpass times (around 05:00 Coordinated Universal Time (UTC) which is equivalent to between 10:30 and 11:30 a.m., Indian Standard Time (IST).

## 3. Results and Discussions

### 3.1. T\*/Fr Spaces

$M_o$ values obtained from the simplified triangle method, plotted in T\*/ Fr space (Figure 2) clearly show a well-defined triangular shape. The estimated soil surface moisture obtained from the third-order polynomial algorithm (Equation (9)) also exhibits the characteristic shape of a triangle. A triangular pattern results from the fact that as Fr increases the T\* decreases, a consequence of the fact that vegetation obscures the highly variable surface dryness (temperature) beneath it while itself maintaining a fairly constant temperature close to that of the air.

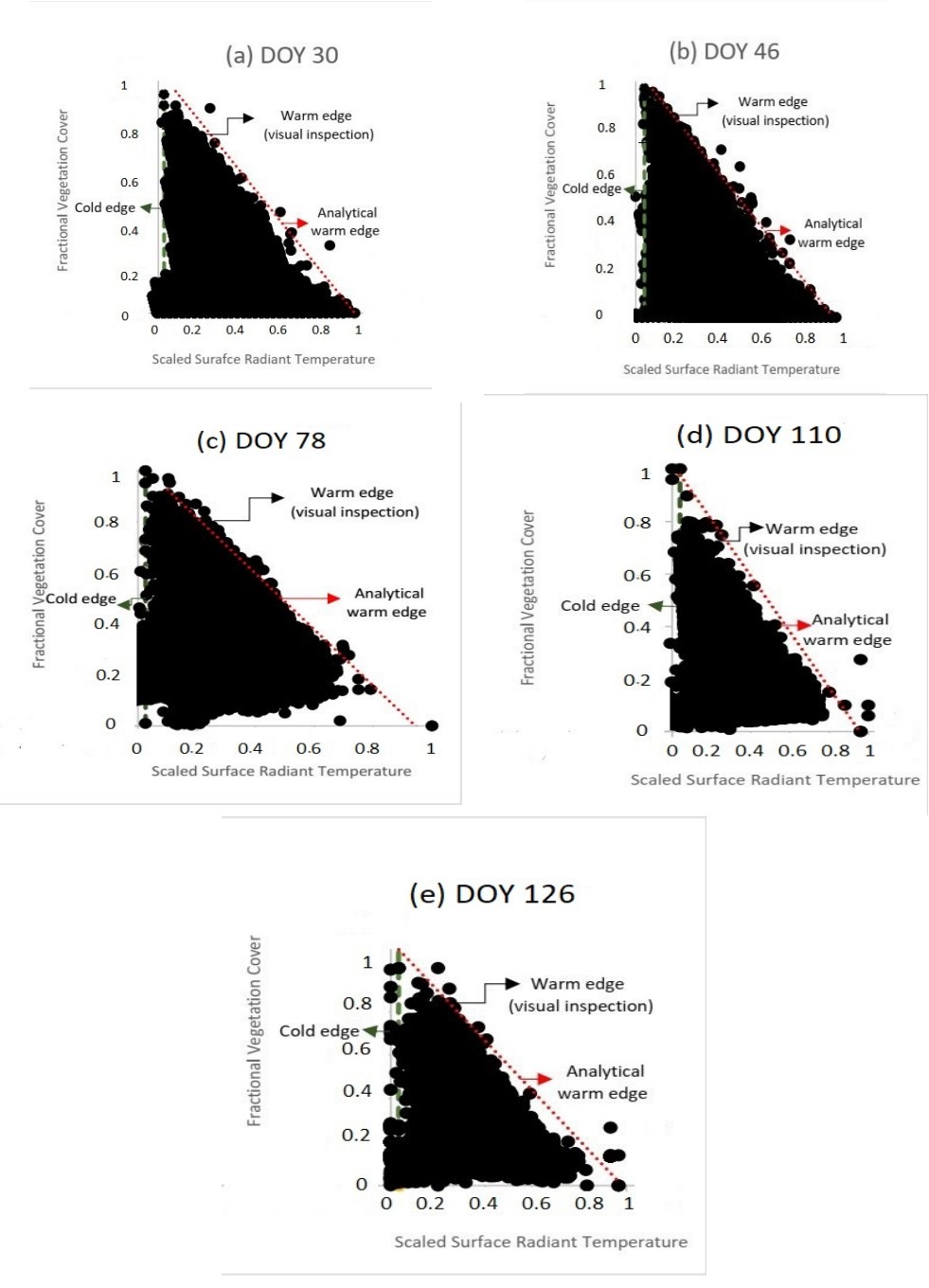

**Figure 2.** Triangles formed in T\*/Fr space for each of the five overpass days (**a–e**), where T\* is the scaled surface radiant temperature and Fr is the fractional vegetation cover.

These figures (Figure 2) agree with the interpretation of T*/Fr space by various authors [6,27–32] showing the salient features of the triangle: the warm and cold edges and the triangular relationship between soil water content and fractional vegetation cover.

The sloping right edge of the distribution, the so-called "warm edge", constitutes the limit of dryness of the soil for a given value of Fr. Pixels tend to show a rather sharp edge, thereby allowing one to easily delineate this feature, which corresponds therefore to a value of $M_o = 0$. Note that this edge does not imply a drying out of the vegetation itself, so that the evapotranspiration fraction EF is not necessarily equal to zero except at the lower right-hand vertex where Fr = 0 and T* = 1.0.

The left edge of the triangle is called the "cold edge". Pixels with cool surface temperature tend to cluster around the edge, which is not as well defined as the warm edge, but nevertheless defines a limit of soil wetness, $M_o = 1.0$. This feature tends to extend vertically from the bare soil line, Fr = 0. By definition EF along this edge is also equal to 1.0 and both EF and $M_o$ are assumed to vary linearly across the domain.

Since the domain of the triangle always varies from 0 to 1.0, successive triangles constructed at different times would all be superposed and congruent. This concept of superposition of triangles constitutes the ''universal'' triangle which allows one to monitor the changes in surface moisture condition of a pixel over several successive days. By locating where a pixel falls in the triangle in a particular day and finding its position in successive days, a trajectory in time is formed which graphically depicts the drying or wetting process.

### 3.2. Spatial and Temporal Variability of Moisture Availability

Pixel measurements (Fr and T*) within each triangle were transformed to moisture availability using the aforementioned algorithms (Equations (9) and (10)). The dimension of each pixel is 30 × 30 m. Moisture availability of the individual pixels then was converted to picture elements, showing the spatial distribution of $M_o$ (Figures 3 and 4). These figures demonstrate the great spatial variability of moisture availability over the study area (Chaka block), while adhering to the triangular configuration.

On the whole, soil surface moisture availability decreases gradually from DOY 46 to DOY 110 owing to the absence of rainfall between the period (Figure 3).

Correspondingly, the simulated soil surface moisture using the geometric model algorithm was also mapped (Figure 4). The distribution also varies in both time and space. The range of soil moisture availability also decreases gradually from DOY 46 to DOY 110 due to the aforementioned dryness. This showed that the results from the two algorithms are consistent.

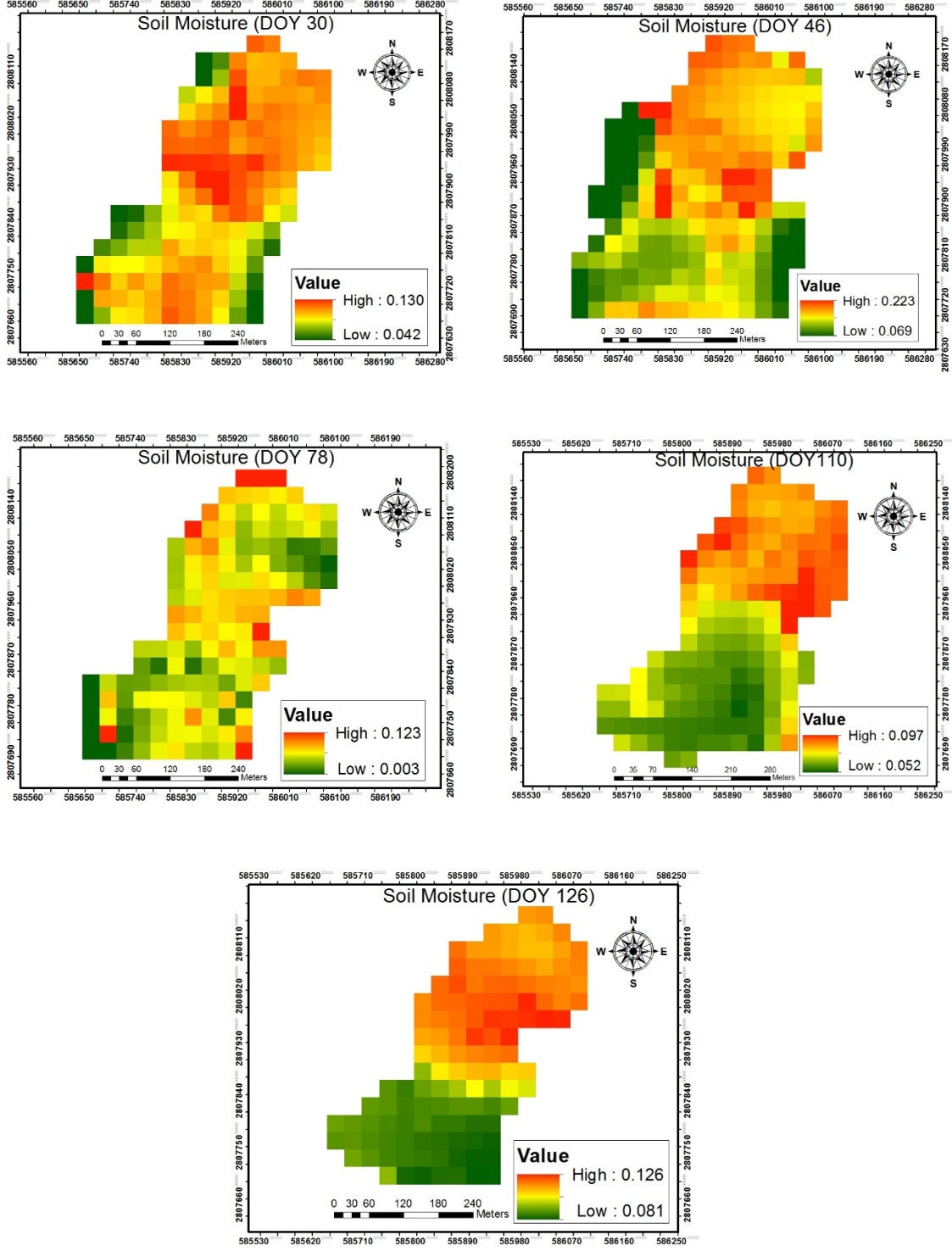

**Figure 3.** Spatial variability of estimated soil surface moisture availability ($M_o$) obtained from the third-order polynomial algorithm in the Chaka block.

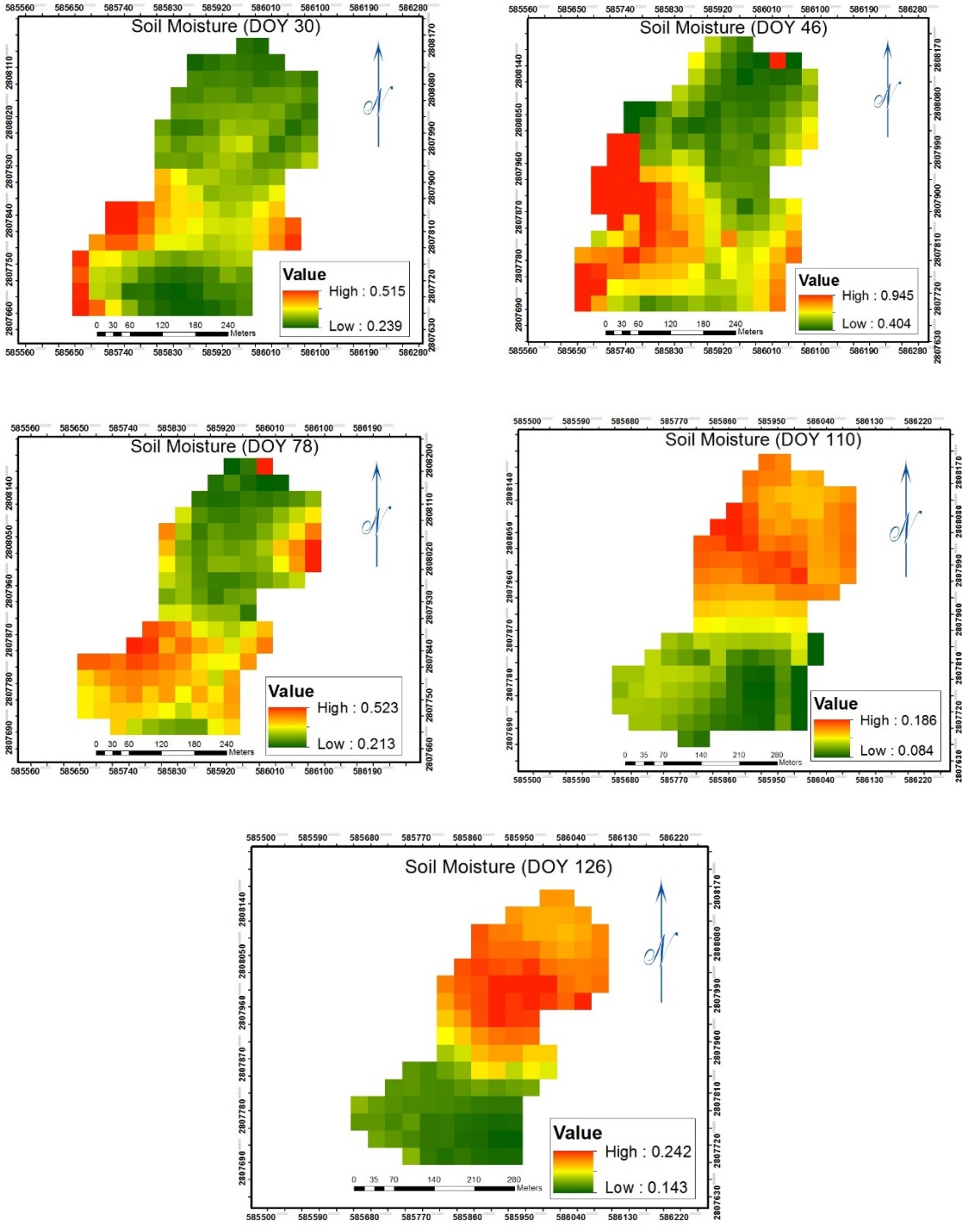

**Figure 4.** Spatial variability of estimated soil surface moisture availability Mo obtained from the simplified triangle method algorithm in the Chaka block.

### 3.3. Soil Water Content Measurements

Soil moisture measurements over 5 and 15 cm depths were obtained using the gravimetric method in the Chaka block for five days on 30 January, 15 February, 19 March, 20 April and 6 May, in 2015. Bulk density of the soil samples was computed to convert gravimetric water content (GWC) to volumetric water content (VWC). These were plotted versus values of $M_o$ produced from the simplified triangle method for three days that had the most reliable measurements: 30 January, 19 March and 20 April in 2015. Implied in Figure 8 for the 5 cm data, the range of $M_o$ from zero to 1.0 corresponds to

soil water content values ranging from 0.04 to just over 0.3 (values calculated by letting x = 0 ($M_o$ = 0) and x =1($M_o$ = 1)).

## 4. Validation

### *4.1. Comparison of $M_o$ between the SVAT Model and the Simplified Triangle Method (Geomtric Model Algorithm)*

The estimated soil surface moisture obtained from the third-order polynomial algorithm and simplified geometric model algorithm were correlated. Figure 5 depicts the $M_o$ values obtained from the two algorithms which correspond to data points where in situ measurements were done. It is apparent from Figure 5 that strong positive correlation was found to exist between the simulated soil surface moisture using the two algorithms. On all the dates, $R^2$ is greater than 0.6 and mostly greater than 0.8. Moreover, the data sets for all days were combined in order to reinvigorate this finding (Figure 6). It was found that the agreement between the two algorithms was promising ($R^2$ 0.7). This result adds to the validation of the use of the recently introduced simplified geometric method by [26].

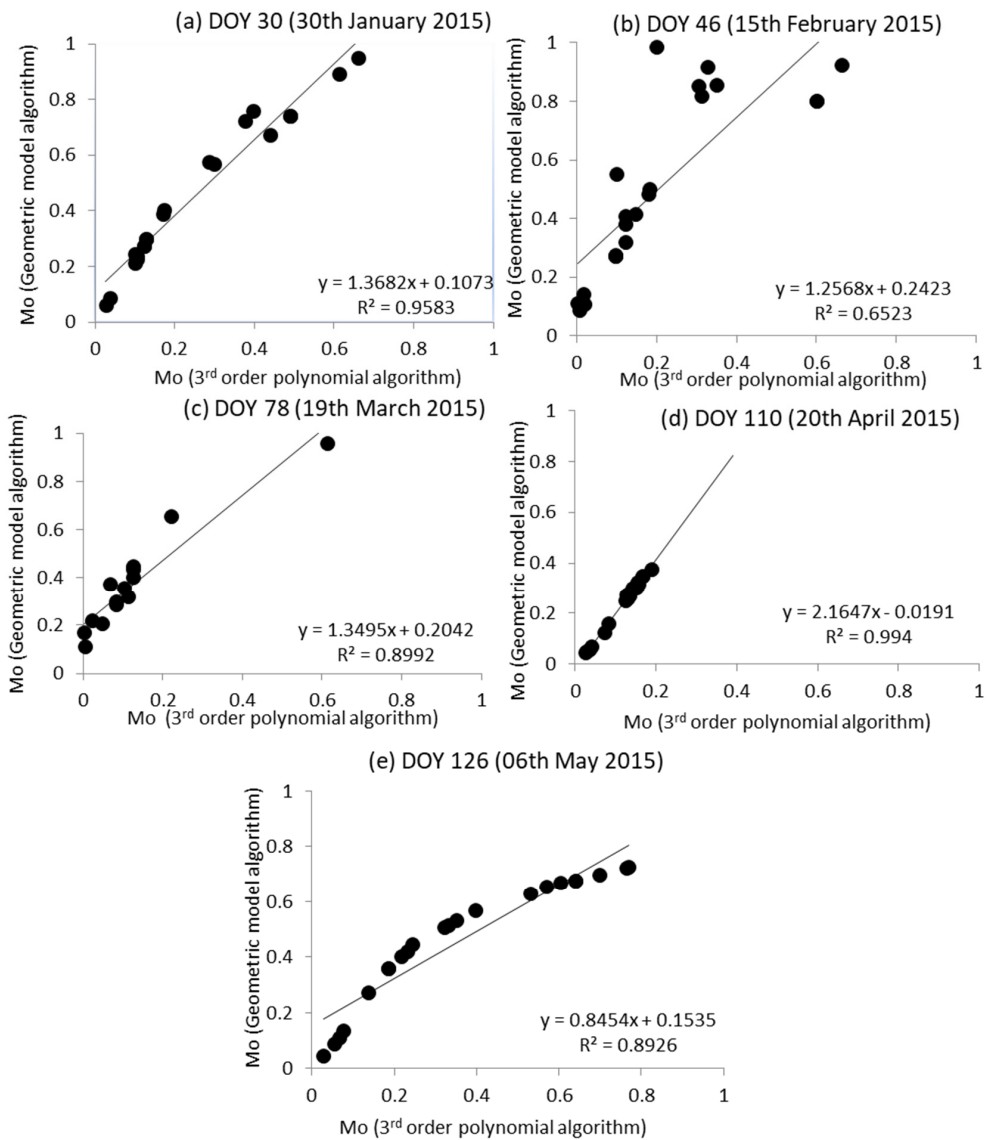

**Figure 5.** Correlation between estimated soil surface moisture availability $M_o$, obtained from a third-order fit to the output of the SVAT model and geometric model algorithm for all five overpass days.

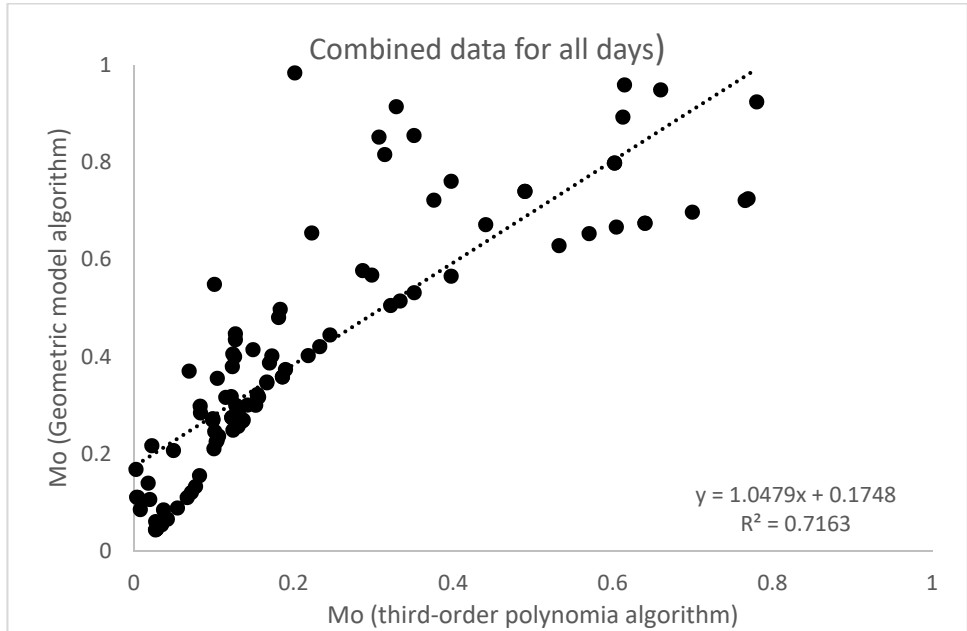

**Figure 6.** Correlation between estimated soil surface moisture availability $M_o$, obtained from a third-order fit to the output of the soil/vegetation/atmosphere/transfer (SVAT) model and geometric model algorithm (combined data for all days).

### 4.2. Comparison of $M_o$ with surface SWC measurements: the limits of validation

Initially these three days (30 January, 19 March and 20 April 2015) showed a small correlation between $M_o$ and the measured soil water content SWC. Capehart and Carlson [3] claim that one reason for the poor correlation between these two variables is the fact that the satellite derived values, in responding to surface radiometric temperature, would correlate best with direct measurements over the top 1 cm of the soil. Consequently, one would not necessarily expect a perfect correlation between the remotely derived $M_o$ and the 0–5 cm soil water content measurements.

It is possible, nevertheless, to demonstrate better agreement between $M_o$ values from the simplified triangle method and the SWC measurements. Besides the mismatch between the surface soil water content represented by $M_o$ and the measured SWC, another source of this poor correlation can be seen graphically on inspection of the triangles. As [16,17] have shown, isopleths of $M_o$ extend from the base of the triangle to its top vertex (Figure 7, modified from [16]), where they merge, thus rendering the values of $M_o$ near the upper vertex indeterminate. Errors in measurement must therefore become increasingly important in going from the base of the triangles to the vertex, at some point completely obscuring the real values of $M_o$.

The rectangles labeled A and B in Figure 7 are referred to here as "error spaces". Irrespective on any model, inherent errors exist in measuring Fr and T*, such as sensor noise, terrain slope, the presence of sub-grid standing water, and clouds, etc. The vertical legs of the two identical rectangles, A and B, shown in this figure (Figure 7), represent this kind of error in measuring Fr, while the horizontal legs of the rectangle correspond to errors in measuring T*. Whatever sizes the reader may wish to assign the sides of these rectangles, it is clear that the error increases as one approaches the upper vertex, at some point implying that the results are completely obscured. (Note, however, that the implied error in evapotranspiration fraction EF hardly changes between rectangles A and B).

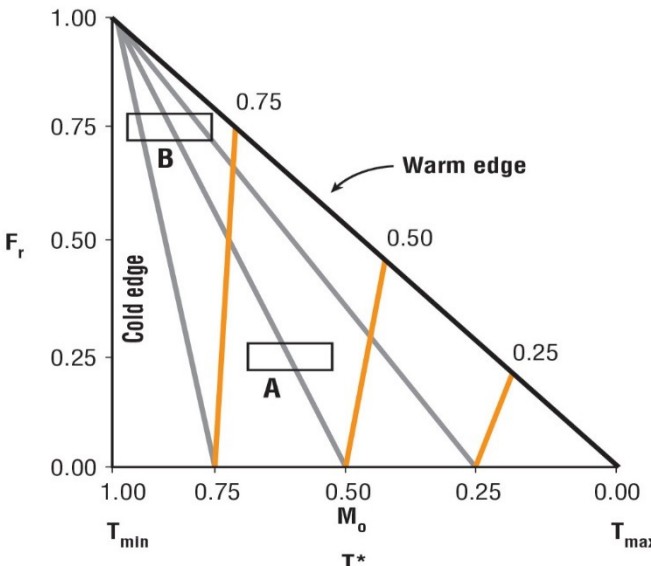

**Figure 7.** Fr versus T* triangle. Isopleths slanting upward toward the left are those of $M_o$ and those sloping upward toward the right are isopleths of evapotranspiration fraction (EF), defined as the ratio of evapotranspiration divided by the net radiation. Identical rectangles A and B represent error spaces, defined in the text; (adapted from [16]).

To show the effect of the error space, we plotted values of $M_o$ obtained from the geometric model algorithm versus SWC only for pixels whose values of Fr were below a certain threshold, while omitting higher vegetation amounts above that threshold. Five values of the threshold were chosen: 0.3, 0.4, 0.5, 0.6 and 0.7 (Table 1).

**Table 1.** R squared and root mean square error values for threshold tests of $M_o$ obtained from the geometric model algorithm versus measured SWC.

| Threshold | No. of Points | R Squared | | RMSE | |
|---|---|---|---|---|---|
| | | 5 cm | 15 cm | 5 cm | 15 cm |
| Threshold 0.3 | 10 | 0.434 | 0.349 | 0.191 | 0.189 |
| Threshold 0.4 | 15 | 0.494 | 0.102 | 0.256 | 0.449 |
| Threshold 0.5 | 26 | 0.338 | 0.325 | 0.231 | 0.229 |
| Threshold 0.6 | 58 | 0.206 | 0.229 | 0.284 | 0.289 |
| Threshold 0.7 | 38 | 0.123 | 0.150 | 0.283 | 0.285 |
| Without threshold | 58 | 0.192 | 0.142 | 0.285 | 0.294 |

A threshold of 0 therefore pertains to only bare soil pixels, while a threshold of 1.0 pertains to the entire image. Table 1 shows that the values of $R^2$ and the RMS error generally improve as the threshold is decreased and less vegetation is included in the sample.

Figure 8 shows the graph of $M_o$ versus SWC for the 0.3 threshold. Values of $R^2$ in Figure 8 would be much higher for the 5 cm measurements were it not for these two outliers for 5 cm measurements near the lower axis. The values of $R^2$ were appreciably higher for thresholds below 0.3, but these graphs are not included because they are based on too few pixels.

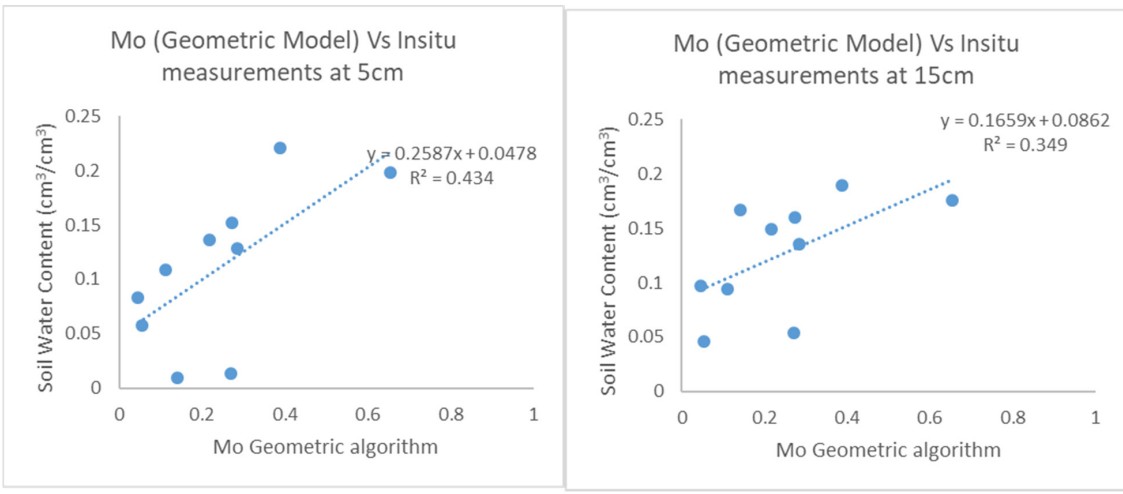

**Figure 8.** Correlation between soil water content and $M_o$ produced from the geometric model algorithm for a threshold of 0.3 (pixels with values of Fr greater than 0.3 were discarded); the left side corresponds to the 0–5 cm depth SWC measurements and the right side is for the 0–15 cm depth measurements.

This relationship from Table 1, plotted in Figure 9, has an $R^2$ of about 0.62 and suggests by extrapolation an $R^2$ value for completely bare soil of about 0.74 (value calculated by letting y = 0 in the regression equation). Similarly, by extrapolation, the $R^2$ would be close to zero for the entire field corresponding to a threshold of 1.0. While these deductions are only speculative, they do support the implication of the error spaces shown in Figure 7. Conversely, however, no such relationship is found in the 15 cm measurements.

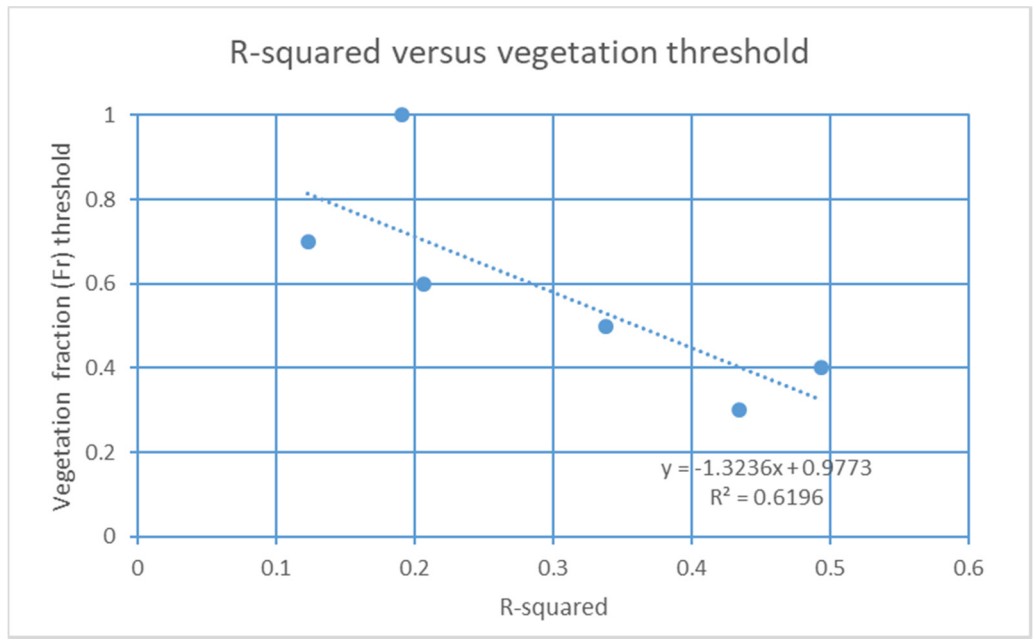

**Figure 9.** Vegetation fraction (Fr) threshold versus the correlation between remotely derived $M_o$ from the simplified triangle method and that measured at the surface in the 0–5 cm layer.

## 5. Conclusions

Although various studies have described the triangle method, (and, more recently, the simplified triangle method) and these show examples of the triangles formed in T*/Fr space, such as in Figure 2., this study provides two new results, including one very important implication for validating remotely determined soil water content, using optical and thermal measurements. These are:

Lack of a high correlation between $M_o$ and soil water content (SWC) found in this and other studies (e.g., [3]) can also be due to the deleterious effects of vegetation as well as the mismatch between SWC measurements made over a 5 cm depth and those obtained remotely using optical and thermal measurements. Correlations systematically improved when limited to smaller and smaller values of fractional vegetation cover. We therefore suggest that validation studies that attempt to assess the fidelity of remotely measured surface soil water content using optical and thermal measurements compared with in situ surface measurements should confine the comparisons to pixels that have little or no vegetation cover. Moreover, we strongly recommend that surface measurements be made over layers with depths as shallow as possible, preferably 5 cm or less.

Close agreement occurred in comparing values of $M_o$ obtained from the simplified triangle method with those from a soil/vegetation/atmosphere/transfer (SVAT) model, thereby lending credence to the simplified triangle method.

In conclusion, the simplified triangle method shows markedly improved agreement between measured SWC and those obtained from this method of optical and thermal remote sensing, when limited to nearly bare soil pixels.

**Author Contributions:** Conceptualization, A.A.K.; funding acquisition, T.N.C.; methodology, A.A.K.; resources, H.S.U.; software, H.S.U.; supervision, T.N.C.; validation, T.N.C.; writing—original draft, A.A.K.; writing—review and editing, A.A.K., T.N.C. and H.S.U. All authors have read and agreed to the published version of the manuscript.

**Funding:** This research received no external funding.

**Acknowledgments:** The authors acknowledge the support of Kano State Government, Federal Republic of Nigeria through Overseas Scholarship Scheme, 2015. Special thanks are extended to the anonymous reviewers and handling editor for their constructive comments.

**Conflicts of Interest:** The authors declare no conflict of interest.

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
