# Peer review of "Limitations in Validating Derived Soil Water Content from Thermal/Optical Measurements Using the Simplified Triangle Method"

_remotesensing, doi:10.3390/rs12071155_

Round 1
Reviewer 1 Report
Please find attached document

Reviewer 2 Report
I congratulate the authors on a well written manuscript.
In comparing Mo from geometric model to 3rd order poly algorithm in section 4.1, the comparison of the 5 different days in Fig 5 is of some interest, but a more accurate estimate of agreement would be to combine the data from all days to determine agreement of the 2 algorithms. What is the R2 when all the data are combined?
Table 1 includes the word "threshold" in the title with no prior introduction to what the threshold is. It would probably be better to insert Table 1 after threshold is defined and Table 1 is referenced in the text.
Table 1 and related text refer to Mo. Would you clarify which Mo this is (Geometric model algorithm or 3rd order poly algorithm)?
I think the terms geometric model algorithm and simplified triangle method are synonyms, but this use of different terms causes confusion. Can you use the same terms?
For even vegetative threshold 0.3, R2 is relatively low and RMSE is high even for 0-5 cm. Acceptability of this high variance to actual data would presumably depend on the how the estimate is used. Can you describe a use where this high variation to actual SWC is practically useful?
The primary conclusions of using the method with little to no vegetation and minimal soil depths are certainly not surprising.
The potentially most useful conclusion is that the simplified triangle method can be used to replace the SVAT model calculation. Assuming R2 from all 5 days together (as requested above) shows good agreement with the SVAT model method, the agreement with actual SWC is still low enough to question whether it is useful. Can you provide a context where this relatively low level of agreement can be useful?
The results would be improved by comparing the agreement of your sensing method to actual SWC to other sensing methods--even if they are not from satellite data--to provide some context of how satellite imagery analysis compares to other state-of-the-art methods.
Reviewer 3 Report
The simplified triangle method was used to derive soil water content on the basis of thermal and optical measurements. This is a well-known approach. However, the authors nicely show the abilities and limitations of the method. This gives a valuable contribution to the state of the art.
The manuscript is well written and organized and of interest for the readers of Remote Sensing.
Only some minor corrections are needed:
Line 71: Use SI units: here meters above sea level instead of feet.
Line 74: Correct to "°C".
Line 75: Suggested to write “... climate according to Köppen [16].”
Line 79-80: Write species names in italics script. Correct to “Butea monosperma”.
Line 95: The correct writing of the unit is “W / (m² sr µm)”. Please correct throughout the manuscript.
Line 100: No need to write “top of atmosphere” and other terms with upper case letters.
Line 103: No need to explain p. (use the Greek lower case letter “pi”).
Line 131: “… temperature was used …”.
Line 137: Pavements are no soil. Change “bare soil, such as surface paving” to “bare surfaces, such as paving”.
Line 149-150: No need to write “The other variables are self-evident”. Please correct throughout the manuscript.
Line 151 “trial and error approaches”. Does this mean using an algorithm?
Fig. 5: Some of the data and especially clearly those in Fig. 5e follow a non-linear curve. The linear fit does not reflect the data.
Line 256: Add a reference number to “modified from Carlson and Petropoulos, 2019”.
Round 2
Reviewer 1 Report
Please find attached file

Reviewer 2 Report
Thanks for doing a good job of responding to questions/concerns.
Author Response
It is our pleasure.